# Sequence-based genetic mapping of *Cynodon dactylon* Pers. reveals new insights into genome evolution in Poaceae

Tilin Fang[1], Hongxu Dong[2], Shuhao Yu[1], Justin Q. Moss [3], Charles H. Fontanier[3], Dennis L. Martin[3], Jinmin Fu[4] & Yanqi Wu [1✉]

Bermudagrass (*Cynodon dactylon* Pers.) is an important warm-season perennial used extensively for turf, forage, soil conservation and remediation worldwide. However, limited genomic information has hindered the application of molecular tools towards understanding genome evolution and in breeding new cultivars. We genotype a first-generation selfed population derived from the tetraploid (4x = 36) 'A12359' using genotyping-by-sequencing. A high-density genetic map of 18 linkage groups (LGs) is constructed with 3,544 markers. Comparative genomic analyses reveal that each of nine homeologous LG pairs of *C. dactylon* corresponds to one of the first nine chromosomes of *Oropetium thomaeum*. Two nested paleo-ancestor chromosome fusions (ρ6-ρ9-ρ6, ρ2-ρ10-ρ2) may have resulted in a 12-to-10 chromosome reduction. A segmental dissemination of the paleo-chromosome ρ12 (ρ1-ρ12-ρ1, ρ6-ρ12-ρ6) leads to the 10-to-9 chromosome reduction in *C. dactylon* genome. The genetic map will assist in an ongoing whole genome sequence assembly and facilitate marker-assisted selection (MAS) in developing new cultivars.

[1] Plant and Soil Sciences Department, Oklahoma State University, Stillwater, OK 74078, USA. [2] Plant and Soil Sciences Department, Mississippi State University, Starkville, MS 39762, USA. [3] Horticulture and Landscape Architecture Department, Oklahoma State University, Stillwater, OK 74078, USA. [4] Coastal Salt Tolerant Grass Engineering and Technology Research Center, Ludong University, Yantai, China. ✉email: yanqi.wu@okstate.edu

Common bermudagrass, *Cynodon dactylon* var. *dactylon* is an economically and ecologically important warm-season perennial grass that has been widely cultivated for turf, forage, soil protection, and remediation in the world[1]. The grass is the only *Cynodon* taxon that enjoys globally remarkable distribution between 45° S. Lat. and 53° N. Lat[2]. and harbors enormous genetic diversity[3,4]. In modern plant breeding programs, *C. dactylon* is often crossed with African bermudagrass (*C. transvaalensis* Burtt-Davy) in generating vegetatively propagated, fine-textured turfgrass cultivars[5] or with *C. nlemfuensis* var. *nlemfuensis* Vanderyst in developing highly digestible and high yielding forage cultivars[6]. Collectively, forage and turf bermudagrass crops have been grown on ~20–25 million hectares in the USA, demonstrating its substantial economic value[7]. Although it is difficult to estimate the acreage of *C. dactylon* grown in many other countries, its use for turf and forage is extensive.

The base chromosome number of *Cynodon* species is x = 9[8]. A series of euploidies including diploid (2x = 18), triploid (3x = 27), tetraploid (4x = 36), pentaploid (5x = 45), and hexaploid (6x = 54) have been reported in the genus with samples collected worldwide over the past century[8–16]. Nevertheless, tetraploid is the predominant ploidy in *C. dactylon* var. *dactylon* and in the *Cynodon* genus in general[2,12–15]. Harlan and de Wet[2] proposed "all tetraploids are autotetraploids" in *C. dactylon* var. *dactylon* and that these probably were derived from the diploid var. *aridus* (2x = 18) on the basis of morphological, cytogenetic, and rhizomatous similarity between the two taxa.

DNA markers are powerful tools that have been employed in *C. dactylon* to investigate genomic structure and constitution, and to construct genetic maps. In one study, an F₁ progeny population derived from *C. dactylon* 'T89' (4x = 36) × *C. transvaalensis* 'T574' (2x = 18) was analyzed to construct two genetic maps, one for each parent, with single-dose restriction fragments[17]. They reported that the *C. dactylon* T89 exhibited "polysomic inheritance of an autotetraploid" based on the predominance of coupling linkages over repulsion. Using the same progeny population, Harris-Shultz et al.[18] observed disomic inheritance for multiple markers on seven of 34 linkage groups (LGs), and therefore reported that T89 may be a segmental allotetraploid or an allotetraploid, rather than an autotetraploid. This population was also used in a recent investigation by Khanal et al.[19] who contributed additional SSR markers to the previous maps. Their study indicated that the *C. dactylon* parent was in a process of diploidization after a whole genome duplication (WGD) event (i.e., an autotetraploid behaving as a segmental polyploid), as some markers showed polysomic inheritance[19]. In a different line of experiments, *C. dactylon* SSR markers showed disomic inheritance in two first-generation selfed (S1) progeny populations[20]. Between the two S1 populations, the one derived from selfing of 'A12359' had less segregation distortion than that derived from 'Zebra'. Consequently, the A12359 population was used for subsequent research including the development of an SSR marker-based genetic map[21]. All SSR markers showed disomic inheritance in the mapping population[21]. Obviously, more research is needed to definitively reveal the genome constitution and subgenome differentiation of *C. dactylon*.

Previous genetic mapping research in *C. dactylon* has generated valuable information, yet a relatively small number of markers, specifically 291 loci in T89[19] and 252 loci in A12359[21], have been mapped. Third-generation marker systems, such as genotyping-by-sequencing (GBS) can generate a large volume of single nucleotide variants (SNVs), which can be used to establish saturated genetic maps. Accordingly, the major objective of the present study was to construct a high-density genetic map using GBS generated SNVs. *C. dactylon* is a member of the genus *Cynodon*, subtribe Eleusininae, tribe Cynodonteae, subfamily Chloridoideae, and family Poaceae[22,23], and the evolution history of *Cynodon* is elusive[24]. As all SNVs in this study were developed from *C. dactylon* DNA, the tagged sequences would provide landmarks for comparative genomic analysis. Therefore, the second objective was to explore the evolution pathway of *C. dactylon* as compared with fully sequenced genomes of selected species in the grass family. The findings would add to the knowledge pool in the evolution of the grass family.

## Results

**GBS and SNV calls.** One hundred and thirty S1 progeny and the parent A12359 were sequenced with GBS. A total of 884,977,554 reads were generated, of which 846,616,373 reads (96%) matched the *Ape*KI enzyme cut site remnant (CWGC) and the 96 barcodes. Sequenced reads were trimmed to 64 bp in length, and a total of 232,166 SNVs were obtained with loose SNV calling criteria (minor allele frequency ≥ 0.01). After removing genotype data with < 6 reads or a missing rate > 10%, a total of 39,013 SNVs were retained for further filtering. A chi-square test was then conducted on each retained SNV to examine the expected segregation ratio of 1:2:1, and the 7443 SNVs meeting this criterion at $P > 0.01$ were retained for genetic mapping.

**Genetic map.** The filtered 7443 SNV markers and 266 SSRs from Guo et al.[21] were used for LG construction. Markers with segregation distortion ($P < 0.01$) were excluded during the framework LG construction. Initially, 18 LGs were made using the maximum likelihood mapping algorithm, of which 16 LGs showed extensive genomic synteny with eight of the 10 *O. thomaeum* chromosomes (Chr. 1–7, 9). The remaining two LGs showed clear synteny with *O. thomaeum* chromosome 8 but with a small number of markers in each LG, resulting in large intermaker spacing. Therefore, to retrieve more markers for these two small LGs, a new dataset was created as follows: markers of these two small LGs were combined with segregation distorted markers ($P < 0.01$) showing significant alignment with *O. thomaeum* chromosome 8 (i.e., comparative genomics approach as described in Methods). Two LGs, 15 and 16, were obtained from this new dataset. After removing redundant markers with a marker interval < 0.4 cM, each LG was recalculated using the regression mapping algorithm. The maximum likelihood algorithm was efficient in ordering markers but often generated a severely expanded genetic map, whereas the regression mapping algorithm was computationally intensive but generated a reasonable map length. Finally, a total of 3544 markers (3322 SNVs and 222 SSRs) were resolved into 18 LGs. The genetic map spanned 1467.3 cM with an average inter-marker spacing of 0.41 cM (Table 1, Fig. 1a). Based on comparative mapping results, extensive genomic collinearity was observed between *C. dactylon* and *O. thomaeum* (Fig. 1b). On average, each of the first nine *O. thomaeum* chromosomes matched two LGs of *C. dactylon*. Therefore, we were able to determine the two homeologous LGs in *C. dactylon* based on their homology to the *O. thomaeum* chromosomes (Table 1). We further numbered *C. dactylon* LGs and designated their orientation based on homology with *O. thomaeum* chromosomes. The number of markers per LG ranged from 91 on LG 16 to 291 on LG 1. The length of each LG ranged from 55.26 cM for LG 16 to 129.95 cM for LG 4. Two genetic gaps (a distance between two adjacent markers larger than 15 cM) were detected on the 18 LGs. They were located on LG 4 with a genetic distance of 20.67 cM and LG 8 with a gap of 34.12 cM.

**Analysis of segregation distorted markers.** All 39,013 SNV markers were initially used to analyze the distribution of segregation distortion. Due to computational constraint, redundant

**Table 1 Statistics summary for the *Cynodon dactylon* linkage groups (LGs) of single nucleotide variants (SNVs) and simple sequence repeats (SSRs) as related to chromosomes of *Oropetium thomaeum*.**

| *Oropetium thomaeum* Chr. No. | *Cynodon dactylon* | | | | | | | | |
|---|---|---|---|---|---|---|---|---|---|
| | LG no. | Genetic length (cM) | Markers no. | Average distance (cM) | SNV Markers no. | SSR Markers no. | Markers no. on distorted LG | Distorted markers no. | Distorted markers ratio |
| Chr 1 | 1 | 100.4 | 291 | 0.35 | 266 | 25 | 251 | 127 | 0.51 |
| | 2 | 94.2 | 224 | 0.42 | 204 | 20 | 174 | 97 | 0.56 |
| Chr 2 | 3 | 63.5 | 221 | 0.29 | 197 | 24 | 156 | 75 | 0.48 |
| | 4 | 129.9 | 215 | 0.60 | 208 | 7 | 207 | 132 | 0.64 |
| Chr 3 | 5 | 113.1 | 165 | 0.69 | 147 | 18 | 266 | 213 | 0.80 |
| | 6 | 84.5 | 120 | 0.70 | 104 | 16 | 250 | 206 | 0.82 |
| Chr 4 | 7 | 86.3 | 242 | 0.36 | 222 | 20 | 219 | 147 | 0.67 |
| | 8 | 69.1 | 113 | 0.61 | 109 | 4 | 42 | 36 | 0.86 |
| Chr 5 | 9 | 81.7 | 198 | 0.41 | 181 | 17 | 120 | 63 | 0.53 |
| | 10 | 64.8 | 161 | 0.40 | 159 | 2 | 104 | 67 | 0.64 |
| Chr 6 | 11 | 75.6 | 220 | 0.34 | 212 | 8 | 179 | 115 | 0.64 |
| | 12 | 72.1 | 248 | 0.29 | 235 | 13 | 168 | 119 | 0.71 |
| Chr 7 | 13 | 73.0 | 254 | 0.29 | 240 | 14 | 129 | 66 | 0.51 |
| | 14 | 82.2 | 260 | 0.32 | 243 | 17 | 142 | 67 | 0.47 |
| Chr 8 | 15 | 79.7 | 141 | 0.57 | 137 | 4 | 141 | 94 | 0.67 |
| | 16 | 55.3 | 91 | 0.61 | 89 | 2 | 92 | 66 | 0.72 |
| Chr 9 | 17 | 71.4 | 191 | 0.37 | 184 | 7 | 127 | 84 | 0.66 |
| | 18 | 70.5 | 189 | 0.37 | 185 | 4 | 137 | 87 | 0.64 |
| Total | | 1467.3 | 3544 | 0.41 | 3322 | 222 | 2904 | 1861 | 0.64 |

markers with genetic distances < 1.5 cM were removed after the first round calculation with the maximum likelihood mapping algorithm, resulting in 2904 markers retained. Using these 2904 markers, of which 1861 (64.08%) were distorted markers, we recalculated the 18 LGs under the regression algorithm (Fig. 2). The number of distorted markers on each LG are given in Table 1. The segregation distorted markers were observed on all 18 LGs but not randomly distributed. It appears that most of the segregation distorted markers formed clusters on each of the LGs (Fig. 2). Approximately 80% and 82% of the total mapped markers on LG 5 and LG 6 were segregation distorted, respectively. The distorted markers on LG 5 expanded the linkage group by ~71 cM compared with that from the non-distorted map (Fig. 2), which was primarily due to their aggregation on one telomeric region (Fig. 2). Similarly, the distorted markers on LG 6 elongated the LG by ~90 cM compared with its counterpart from the non-distorted map, and these markers mostly appeared to intermix with the non-distorted markers. Comparing the LGs with and without distorted markers, we observed that the distorted markers did not dramatically change the marker orders on the LGs.

**Comparative mapping.** Among the three Chloridoideae species *C. dactylon*, *O. thomaeum,* and *Zoysia japonica*, it is evident that a high level of chromosome-level collinearity exists between *C. dactylon* and *O. thomaeum* (Fig. 1b) and that a similar level of collinearity exists between *Z. japonica* and *C. dactylon* (Supplementary Fig. 3A). The results are not unexpected since *O. thomaeum* and *C. dactylon* are classified into the same tribe Cynodonteae[23]. *O. thomaeum* ($2x = 20$) is an emerging model for desiccation tolerance and genome size evolution in grasses and has an extensive degree of chromosome-level collinearity with *Sorghum bicolor* ($2x = 20$)[25,26]. Among the 3322 SNVs mapped on this *C. dactylon* genetic map, 702 SNV tagged sequences (21.13%) shared high synteny with *O. thomaeum* genome (Table 2). A clear 2:1 chromosome correspondence ratio was observed between the 18 LGs of *C. dactylon* and the chromosomes 1–9 of *O. thomaeum*. However, no discernable chromosome-level correspondence between *C. dactylon* LGs and *O. thomaeum* chromosome 10 was found (Fig. 1b). The comparative analysis also showed frequent chromosomal rearrangements in *C. dactylon* relative to *O. thomaeum*. Obvious gaps were observed at centromeric regions for each homeologous LG pair.

Interestingly, inverted insertions of ancestral chromosomal segments were observed in several *C. dactylon* LGs (e.g., LGs 3, 15, 16, 17, 18), and *C. dactylon* LG 6 exhibited syntenic relationship with the long arm of *O. thomaeum* chromosome 3, whereas LG 8 showed synteny with telomeric regions of *O. thomaeum* chromosome 4 in an inverted fashion. *C. dactylon* LGs 15 and 16 were obtained after incorporating segregation distorted markers ($P < 0.01$) and showed less pronounced genomic synteny with *O. thomaeum* chromosome 8.

In order to identify the possible reason for the absence of genomic synteny between *C. dactylon* LGs and *O. thomaeum* chromosome 10, we further investigated all raw SNV tagged sequences that aligned to *O. thomaeum* chromosome 10 and constructed genetic maps using these SNVs. A total of 2048 SNV-tagged sequences were aligned to *O. thomaeum* chromosome 10, of which 320 SNVs were successfully grouped into four new LGs (Supplementary Data 2). Among these 320 mapped SNVs, 32 were mapped on the segregation distorted map, including 7 on LG 3, 6 on LG 4, 6 on LG 5, and 13 on LG 6 (Fig. 2). Thus, these four LGs were named as LG 3-1, LG 4-1, LG 5-1, and LG 6-1, accordingly. When these four LGs were compared with *O. thomaeum* chromosome 10, they showed two homeologous groups, with LG 3-1 and LG 4-1 corresponding to one arm of *O. thomaeum* chromosome 10, and LG 5-1 and LG 6-1 corresponding to the other arm (Fig. 3). As expected, a high degree of chromosome-level collinearity was observed between *Z. japonica* and *C. dactylon* (Supplementary Fig. 3A). The basic chromosome number of *C. dactylon* is 9, while it is 10 in *Z. japonica*. Each *C. dactylon* LG pair corresponded to one pair of *Z. japonica* chromosomes except for *Z. japonica* chromosomes 15 and 16 (Supplementary Fig. 3A). Such mutual correspondence implied their divergent speciation from two common ancestors as *Z. japonica* is an allotetraploid species[27]. Similarly to those observed between *C. dactylon* and *O. thomaeum*, numerous chromosomal rearrangements differentiate these two species. Among 3322 SNVs in the *C. dactylon* genetic map, 703 (21.16%) were unambiguously aligned to *Z. japonica* genome sequence (Table 2).

Similarly, syntenic blocks and collinearity were also observed between *C. dactylon* and the selected grass species in other subfamilies (Fig. 4, Supplementary Fig. 3A–C). Panicoideae is the sister subfamily of Chloridoideae. Three Panicoideae grasses, *S. bicolor* ($2x = 20$), *Setaria italica* ($2x = 18$), and *Miscanthus*

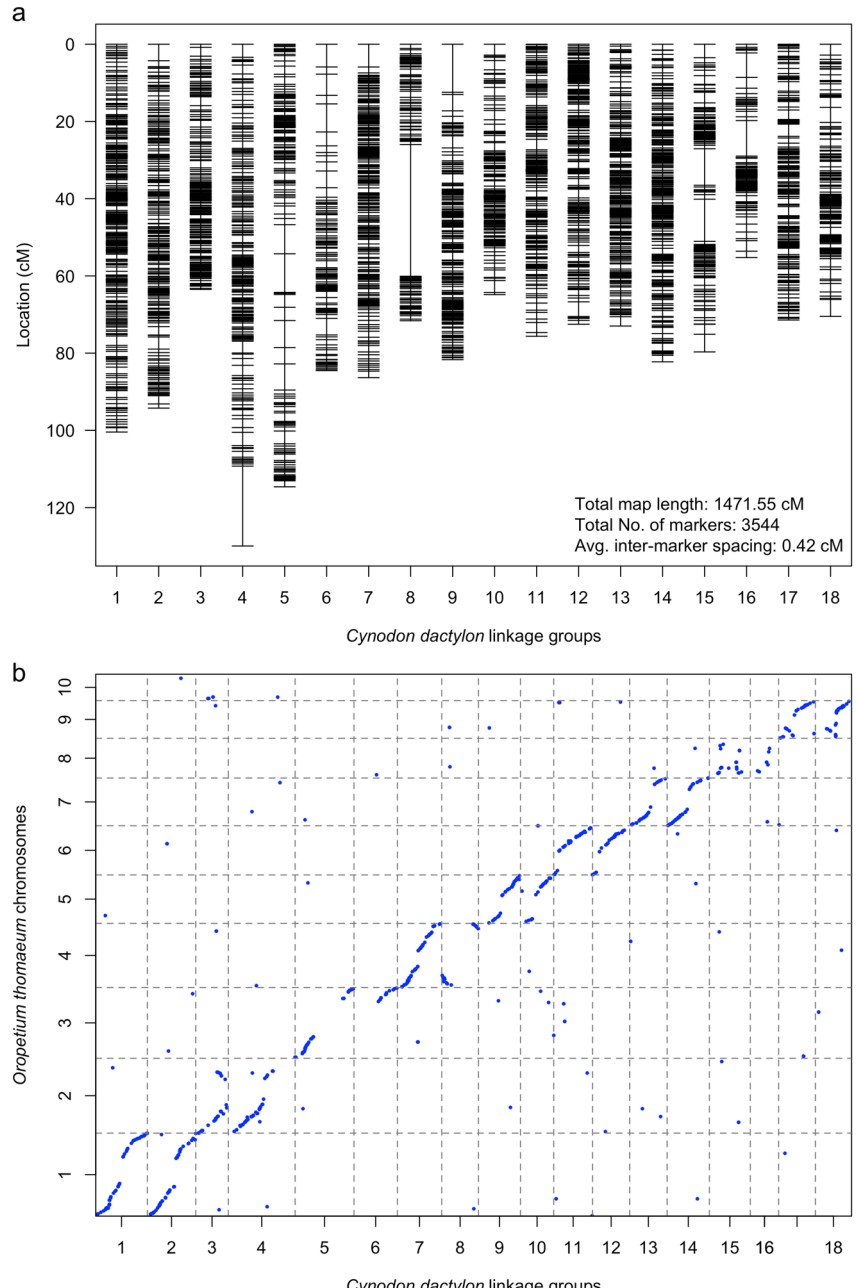

**Fig. 1 Genetic map of common bermudagrass (*Cynodon dactylon*) and its comparison with *O. thomaeum*. a** A high-density genetic map of common bermudagrass 'A12359'. Genetic distance is shown on the left in centimorgans (cM) and linkage group (LG) numbers are shown at the bottom. **b** Genomic synteny plot between *C. dactylon* and *Oropetium thomaeum*. Horizontal axis shows genetic position of markers on 18 *C. dactylon* LGs; vertical axis shows the map position of markers aligned to the 10 *O. thomaeum* chromosomes. Each dot represents a single marker.

*sinensis* (2x = 38) were employed in the comparative analysis (Supplementary Fig. 3B–D). In brief, each *S. italica* chromosome corresponded to two *C. dactylon* LGs (Supplementary Fig. 3B), and each *S. bicolor* chromosome corresponded to two *C. dactylon* LGs except for chromosome 8 (Supplementary Fig. 3C). *M. sinensis* has one more haploid chromosome than *C. dactylon*, and the synteny analysis revealed mutual correspondence between these two species as that observed between *C. dactylon* and *Z. japonica* (Supplementary Fig. 3A, D).

*Oryza sativa* (2x = 24) is a member of Ehrhartoideae (syn. Oryzoideae), representing the phylogenetically most distant species with *C. dactylon* as compared with the other five species in this study. Nine of the 12 *O. sativa* haploid chromosomes

showed 1:2 correspondence with *C. dactylon* 18 LGs except for *O. sativa* chromosomes 9, 10, and 12 (Fig. 4a). The *O. sativa* chromosome 9 showed local synteny with the central part of *C. dactylon* LGs 3 and 4. In similar fashion, the *O. sativa* chromosome 10 was related to the central part of *C. dactylon* LGs 1 and 2. The *O. sativa* chromosome 12, however, did not exhibit any significant connection to *C. dactylon* LGs. In summary, between *C. dactylon* and the six other grass species, highest syntenic levels were observed between *C. dactylon* and the other two Chloridoideae grasses *O. thomaeum* (21.13%) and *Z. japonica* (21.16%; Table 2), followed by three Panicoideae grasses *S. italica* (13.96%), *S. bicolor* (12.34%), *M. sinensis* (12.01%), and tailed by *O. sativa* (8.31%).

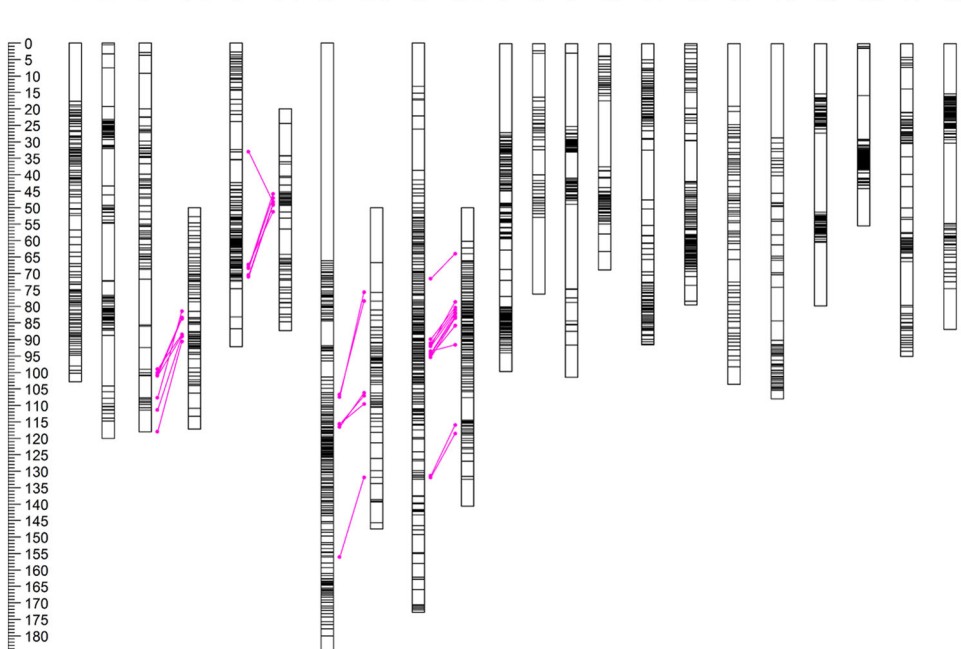

**Fig. 2 Distribution of segregation distortion markers on *Cynodon dactylon* linkage groups and its alignment to *Oropetium thomaeum* chromosome 10.** Segregation distortion markers on linkage groups (LGs) 1–18. 3-1, 4-1 5-1, and 6-1 are the LGs mapped with the SNV markers aligned to *Oropetium thomaeum* chromosome 10. The numbers on the top row indicates the LGs and left ruler shows the genetic distance in centimorgans (cM). Magenta dots aligned with red lines indicates the same markers mapped on both the LGs.

**Table 2 Comparative genomics analysis between *Cynodon dactylon* and six other species in the Poaceae family.**

| Species | Subfamily | Clade | No. of aligned SNV tagged sequences | Alignment rate (%)[a] |
|---|---|---|---|---|
| *Oropetium thomaeum* | Chloridoideae | PACMAD | 702 | 21.13 |
| *Zoysia japonica* | Chloridoideae | PACMAD | 703 | 21.16 |
| *Setaria italica* | Panicoideae | PACMAD | 464 | 13.96 |
| *Sorghum bicolor* | Panicoideae | PACMAD | 410 | 12.34 |
| *Miscanthus sinensis* | Panicoideae | PACMAD | 399 | 12.01 |
| *Oryza sativa* | Ehrhartoideae | BEP | 276 | 8.31 |

*PACMAD* Panicoideae, Arundinoideae, Chloridoideae, Micrairoideae, Aristidoideae and Danthonioideae, *BEP* Bambusoideae, Ehrhartoideae (formerly Oryzoideae) and Pooideae.
[a]Alignment rate was calculated by dividing the number of SNV tagged sequences aligned to a specific genome over the total 3322 mapped SNVs in *C. dactylon* genetic map.

## Discussion

To the best of our knowledge, we developed the most-dense genetic map for *C. dactylon* using SNVs generated through GBS. Compared with the tetraploid *C. dactylon* maps of Khanal et al.[19] and Guo et al.[21] this map has increased marker density by more than tenfold. The average inter-marker spacing in this map was 0.41 cM, while those values from Khanal et al.[19] and Guo et al.[21] were 12.5 cM and 4.3 cM, respectively. The tag sequences of mapped markers in this study have been used to facilitate a whole-genome assembly of *C. dactylon* by resolving uncertainties in assembling large scaffolds due to high heterozygosity in the genotype A12359 (Y.Q.W. and J.F., personal communication). The tag sequences of this high-density genetic map also provided valuable community resources for marker-assisted selection (MAS) and genetic diversity studies.

Segregation distortion is a common phenomenon in plants and animals. In this study, we had to include severely segregation distorted markers ($P < 0.01$) to obtain LGs 15 and 16. Similarly in *Miscanthus*, it was necessary to include segregation distorted markers to obtain the 'missing' LG 15[28]. *Cynodon dactylon* is largely an outcrossing species[29], whereas some genotypes such as A12359 can produce selfed progeny[20]. Although this self-

compatible parent produced hundreds of S1 progeny, many gametes with low fitness or carrying potential lethal alleles in homozygous loci might have perished, and thus were not represented in this S1 population, leading to varying degrees of segregation distortion. Since this population was derived from selfing a single parent, the segregation distortion observed in this study would be more extensive than populations derived from crossing two parents.

The interest concerning the genome of *Cynodon* species started in the early 1930s when *C. dactylon* was recognized as an important turf and forage species[30]. Investigations indicated that tetraploid *C. dactylon* var. *dactylon* is the only taxon of the genus, having worldwide geographic distribution[11]. The cosmopolitan form has contributed the most germplasm/genes to developing modern forage and turf cultivars. One fundamental question regarding the worldwide tetraploid *C. dactylon* genome is whether it is an autopolyploid with polysomic inheritance vs. an allopolyploid with disomic inheritance. Chromosome pairing behavior in meiosis may have provided insights into chromosome affinity. Forbes and Burton[8] reported that *dactylon* cv. Coastal had 0.15 I's, 16.00 II's, and 0.96 IV's and that PI 226011 had 1.36 I's, 15.69 II's, 0.18 III's, and 0.68 IV's. Hanna and Burton[31] observed

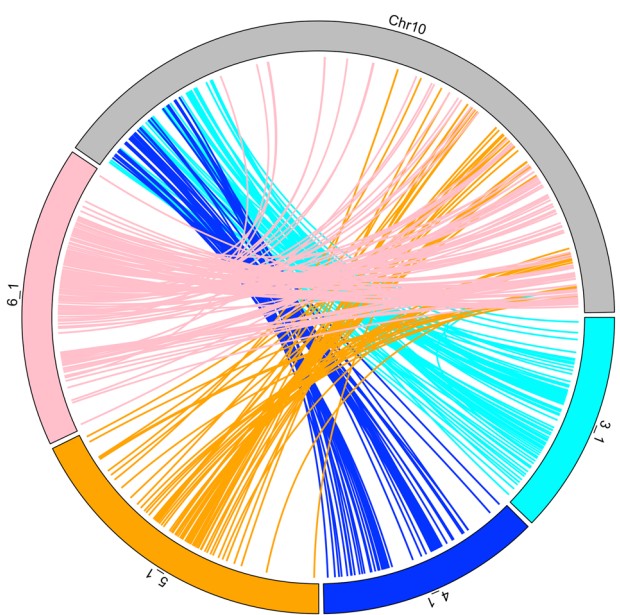

**Fig. 3 Comparison between** *Oropetium thomaeum* **chromosome 10 and four segmental** *Cynodon dactylon* **linkage groups (LGs).** A total of 320 SNVs that aligned to *O. thomaeum* chromosome 10 were investigated and formed four LGs of *Cynodon dactylon*.

similar 0.1–1.72 I's, 16.13–17.45 II's, 0–0.15 III's and 0.26–0.39 IV's in three tetraploid *dactylon* cultivars (Coastal, Midland, and Suwanee). The results may suggest higher affinity/homology between two chromosome pairs, resulting in the tetravalent formation in meiosis. Harlan and de Wet[2] performed a large investigation of meiotic chromosome behavior in *C. dactylon* var. *dactylon* germplasm, including all three races (tropical, temperate, and seleucidus) collected from multiple countries. Their study indicated that among 50 hybrids, 11 had regular meiosis with 18 II's; 13 showed slightly irregular meiosis with 0–2 I's, 17–18 II's, and 0–1 IV's; 19 exhibited irregular meiosis with 2–6 I's, 14–18 II's and 0–2 VI's; and seven demonstrated very irregular meiotic behavior with 0–8 I's, 10–18 II's, and 0–2 VI's. The regular meiotic events were observed in hybrids between parental plants collected from the most similar origins, while the most irregular ones were found in progeny of parents from more distant locations with a few exceptions[2]. It appears that tetraploid *C. dactylon* has a complex genome with various homologies between two subgenomes, which may have evolved over time under different conditions through natural selection.

Guo et al.[20] studied segregation patterns of SSR markers in two S1 progeny populations and reported that the two parental *C. dactylon* plants were allotetraploidy. Using one of the two populations, they mapped 249 SSRs into 18 LGs. Among the mapped markers, 154 non-distorted loci had the 1:1 ratio of coupling vs. repulsion, suggesting the allopolyploid origin of the parent A12359[21]. Roodt and Spies[32] reported that one tetraploid *C. dactylon* was allotetraploidy while another was segmental allotetraploidy. On the basis of observing two strong and two weak signals of 45S rDNA and 5S rDNA FISH probes on metaphase chromosomes in mitosis and 18 bivalents of chromosome pairing in meiosis of one tetraploid *C. dactylon*, Gong et al.[33] indicated that the tetraploid *C. dactylon* had two different subgenomes. The findings in this study lend strong evidence for disomic inheritance and allotetraploid in *C. dactylon*. Since *C. dactylon* is adapted to a large range of environments in the world, it is likely that in the evolutionary process, accumulation of genomic changes under local natural selection pressures has

resulted in the complexity of genome structure and constitution in the species.

Genomic synteny and collinearity is a common feature in the grass family (Poaceae), which is unarguably the most important plant family with about 12,000 species that are classified into 12 subfamilies[23,34]. In the present study, we compared *C. dactylon* with *O. thomaeum* and *Z. japonica* of the subfamily Chloridoideae; *S. bicolor*, *S. italica*, and *M. sinensis* of Panicoideae; and *O. sativa* of Ehrhartoideae (Table 2). In addition, both Chloridoideae and Panicoideae belong to the 'PACMAD' clade (Panicoideae, Arundinoideae, Chloridoideae, Micrairoideae, Aristidoideae and Danthonioideae), whereas Ehrhartoideae belongs to the 'BEP' clade [Bambusoideae, Ehrhartoideae (formerly Oryzoideae) and Pooideae]. Therefore, it is not unexpected that *C. dactylon* showed high genomic syntenic relationships with *O. thomaeum* and *Z. japonica* due to their close phylogenetic relationship within the Chloridoideae subfamily (Table 2). The levels of genomic synteny between *C. dactylon* and the three Panicoideae species were similar, and the results agreed with the relatively close phylogentic relationship between the subfamilies Chloridoideae and Panicoideae. The lowest level of genomic synteny between *C. dactylon* and *O. sativa* further agreed with the most distant phylogenetic relationship between subfamilies Chloridoideae and Ehrhartoideae.

Chloridoideae subfamily is yet to be explored in genomics and evolutionary studies[23]. Most comparative genomic analyses between Chloridoideae grasses have used *S. bicolor*, a well sequenced and studied Panicoideae grass, as reference. Sorghum is in the Panicoideae subfamily, which may have diverged from the common ancestor of Chloridoideae ~31 million years ago. Despite this divergence, VanBuren et al.[26] demonstrated the ten chromosomes in *O. thomaeum* are largely collinear to the corresponding ten chromosomes in *S. bicolor*, though large-scale inversions and translocations were identified. Wang et al.[35] reported WGD in *Z. japonica* relative to *S. bicolor*. In this study, we also observed that each *S. bicolor* chromosome except for chromosome 8 showed collinear synteny with two *C. dactylon* LGs (Supplementary Fig. 3C). Within the Panicoideae subfamily, multiple studies have confirmed the WGD in *Miscanthus* relative to *S. bicolor*[36–38].

It has been proposed that the common ancestor of grasses most likely had a basic chromosome number of seven[39,40]. The basic chromosome number increased to 14 via one WGD, and then reduced to 12 via two nested chromosome fusions (NCFs)[41]. *O. sativa* genome has a basic chromosome of 12 and has been believed to likely resemble the 12 paleo-ancestor chromosomes (ρ) and provides an important reference for comparative genomics studies[35]. In this study, the comparative genomic analysis between *C. dactylon* and *O. sativa* revealed paleo-ancestor chromosome 9 (ρ9) inserted into the centromeric region of paleo-ancestor chromosome 6 (ρ6) to form *C. dactylon* LGs 3 and 4 (ρ6-ρ9-ρ6; Fig. 3b); and paleo-ancestor chromosome 10 (ρ10) inserted into the centromeric region of paleo-ancestor chromosome 2 (ρ2) to form *C. dactylon* LGs 1 and 2 (ρ2-ρ10-ρ2). Such paleo-chromosome merges during speciation have been demonstrated in many other grass species. In *Z. japonica* ($x = 10$), ρ6-ρ9-ρ6 and ρ2-ρ10-ρ2 have been proved to be involved in the 12-to-10 process[35]; in *Eleusine coracana*, ρ6-ρ9-ρ6, ρ2-ρ10-ρ2, and ρ5-ρ12-ρ5 have been reported to be engaged in the 12-to-9 process[42]; in *S. bicolor*, ρ7-ρ9-ρ7 and ρ3-ρ10-ρ3 were demonstrated in the 12-to-10 process[43]; and in *S. italica*, ρ7-ρ9-ρ7, ρ3-ρ10-ρ3, and ρ5-ρ12-ρ5 were demonstrated in the 12-to-9 process[44]. In the Chloridoideae subfamily, all three species *E. coracana*, *C. dactylon*, and *Z. japonica* have undergone ρ6-ρ9-ρ6 and ρ2-ρ10-ρ2 paleo-chromosome merges. With this high-density genetic map, we discovered the tetraploid *C. dactylon* had the

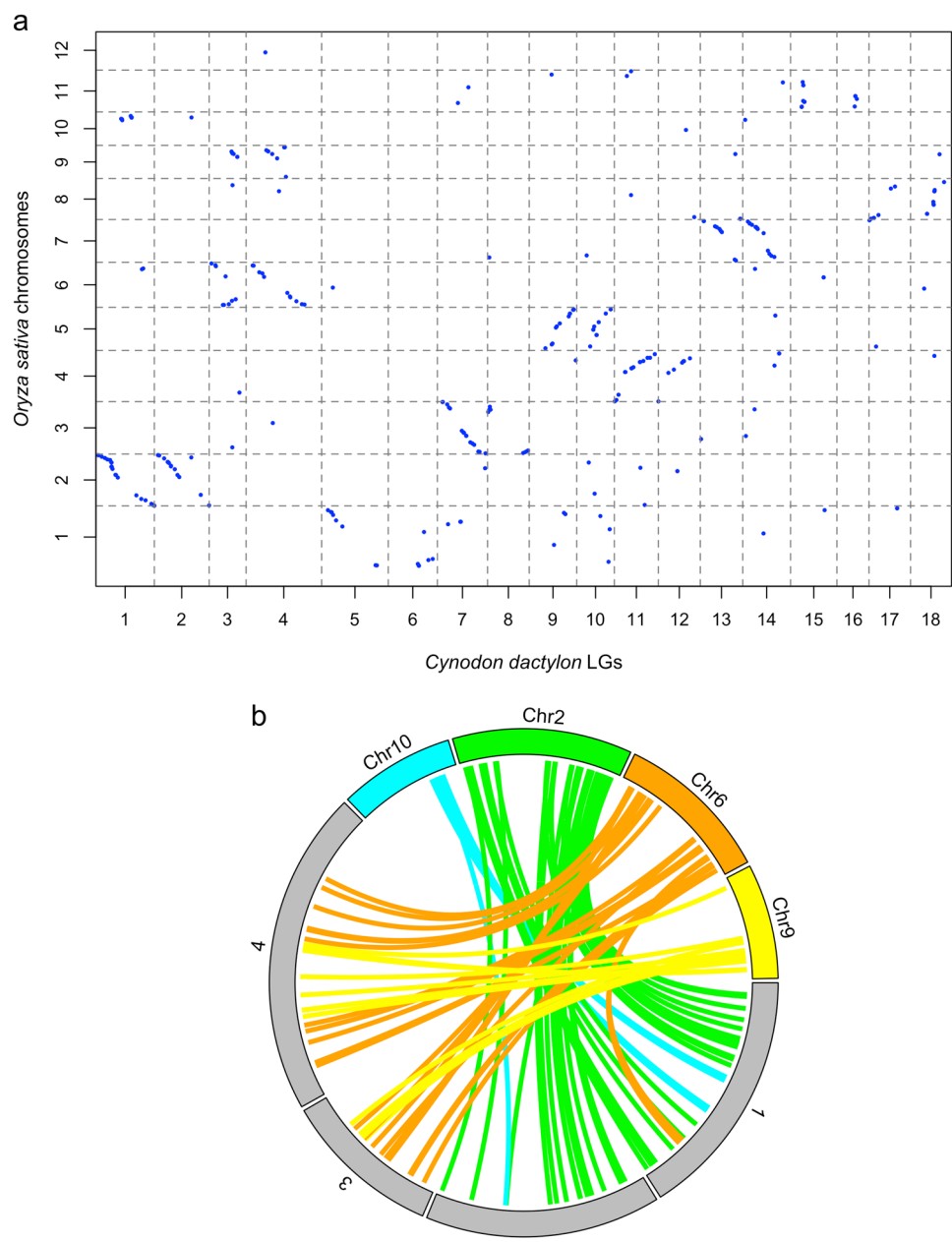

**Fig. 4 Comparison between *Cynodon dactylon* and *Oryza sativa* genomes. a** Comparison between *C. dactylon* linkage groups (LGs) and *O. sativa* chromosomes. **b** Circos plot showing *O. sativa* chromosome 9 inserted into the centromeric region of chromosome 6 to form *C. dactylon* LGs 3 and 4; *O. sativa* chromosome 10 inserted into the centromeric region of chromosome 2 to form *C. dactylon* LGs 1 and 2. The gray filled segments represent *C. dactylon* LGs and color filled segments represent *O. sativa* chromosomes.

same two merges ($\rho6$-$\rho9$-$\rho6$ and $\rho2$-$\rho10$-$\rho2$), resulting in chromosome number reduction from 12 to 10.

An interesting finding from this study was that no genomic synteny was found between *C. dactylon* LGs and *O. thomaeum* chromosome 10, and between *C. dactylon* LGs and *O. sativa* chromosome 12 in the initial analysis (Figs. 2, 4). Both *O. thomaeum* and *O. sativa* have high-quality genome sequences[25,44]. The comparison between *O. thomaeum* and *O. sativa* genome sequences revealed that *O. thomaeum* chromosome 10 is largely syntenic with *O. sativa* chromosome 12 (Supplementary Fig. 3E). This phenonmenon echoes our findings that no genomic correspondence was found between *C. dactylon* LGs with *O. thomaeum* chromosome 10 and *O. sativa* chromosome 12 (Figs. 1, 4). Evolutionary studies suggested that the ancestors of all extant diploid angiosperm species underwent whole-genome duplication, which

is consistent with the progenitors of extant diploid plants experiencing polyploidization followed by diploidization[45–48]. Diploidization includes loss of entire chromosomes or large segments of chromosomes, losses of one copy of duplicated genes, and various chromosome rearrangements. However, the loss of a whole chromosome corresponding to the paleo-ancestor chromosome 12 and *O. thomaeum* chromosome 10 is unlikely to have occurrd during *C. dactylon* genome evolution. The four new LGs (LG 3–1, LG 4-1, LG 5-1, and LG 6-1 in Fig. 2) mapped with the SNVs aligned to *O. thomaeum* chromosome 10 (Fig. 3) indicated that *O. thomaeum* chromosome 10 was divided into two parts, and segmentally disseminated into two *C. dactylon* homoleogous chromosome pairs. LGs 3 & 4 are one pair and LGs 5 & 6 another. This result indicated that the chromosome reduction from 10-to-9 may have happened before the formation of the allotetraploid *C. dactylon*.

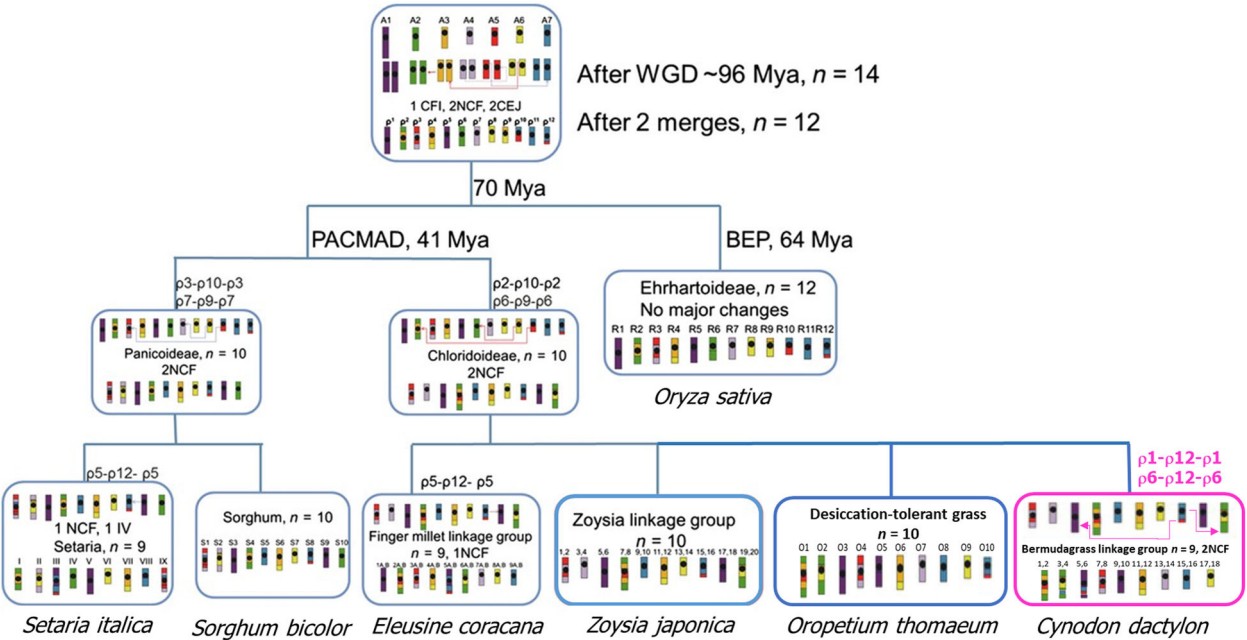

**Fig. 5 Evolutionary model for bermudagrass (*Cynodon dactylon*).** Bermudagrass (highlighted in magenta color) has unique chromosome rearrangements as compared with sequenced and fine-mapped major grasses in the Panicoideae, Chloridoideae, and Ehrhartoideae subfamilies. Chromosomes in rearrangement events are linked with lines. NCF nested chromosome fusion, IV inversion, CFI chromosome fission, CEJ chromosome end–end joining, Mya millions of years ago, WGD whole-genome duplication. Modified from previous publications[35,49].

The ρ6-ρ9-ρ6 and ρ2-ρ10-ρ2 paleo-chromosome rearrangements are common in the Chloridoideae subfamily. The composite structures of bermudagrass LGs 3, 4 (one half of paleo-ancestor chromosome 12 inserted into 6, ρ6-ρ12-ρ6) and LGs 5, 6 (the other half of paleo-ancestor chromosome 12 into 1, ρ1-ρ12-ρ1) are unique to *C. dactylon*. The new findings in the current study represented a different evolution event in *C. dactylon* from the finger millet ρ5-ρ12-ρ5 chromosome rearrangement within the same subfamily[41]. To summarize the results in a phylogenetic context, the evolutionary model to demonstrate the chromosome number reduction for *C. dactylon* was modified from previous publications[35,49] of sequenced and mapped grasses in the Panicoideae, Chloridoideae, and Ehrhardoideae subfamilies (Fig. 5). Evidently, the paleo-chromosome 12 has played an important role in evolution within Chloridoideae. The high-density genetic map developed in this study provides a clear view of the allotetraploid genome structure of *C. dactylon*. The base number of nine chromosomes of the species was demonstrated to be derived from an ancestor of 10 chromosomes through the two chromosomal rearrangement events (ρ1-ρ12-ρ1 and ρ6-ρ12-ρ6). The study confirmed that two NCFs (ρ6-ρ9-ρ6 and ρ2-ρ10-ρ2) happened to all the species tested in the Chloridoideae subfamily. As whole genome sequences are available in the *Cynodon* genus, more in the Chloridoideae subfamily and Poaceae family, further investigations might unveil mechanisms in chromosome evolution. The high-resolution genetic map will be a powerful tool for establishing associations between phenotypes and genotypes, and performing MASs in breeding new cultivars of turf and forage bermudagrass.

## Methods

**Cynodon dactylon mapping population.** A first-generation S1 population derived from the *C. dactylon* genotype A12359 (2n = 4x = 36) was used in this study[13,20]. Detailed information of the population development was described by Guo et al.[21]. In brief, 130 randomly selected individuals from the S1 population were employed for GBS and genetic map construction.

**DNA extraction, library construction, and genotyping by sequencing**. Young leaf samples of the progeny and parent were sent to the Genomic Diversity Facility at Cornell University, Ithaca, NY, where DNA extraction, library preparation, and sequencing were performed. Briefly, genomic DNA was extracted with Quick-DNA™ Plant/Seed 96 Kit (Zymo Research, Irvine, CA), and GBS libraries were prepared using the *Ape*KI enzyme system[50]. In order to improve read depth and reduce missing data, DNA was replicated twice for each progeny and 14 times for the parent, which was distributed in all libraries. A total of three GBS libraries were sequenced on a HiSeq 2500 (Illumina, San Diego, CA) with 100 bp single-end reads.

**Single nucleotide variant calls**. Raw sequence data were investigated for base quality using FastQC v0.11.7 (https://www.bioinformatics.babraham.ac.uk/projects/fastqc/). All three GBS libraries were sequenced with high-quality reads, and only the two ends of reads in each library showed slightly lower quality score (Supplementary Fig. 1). For SNV calling, the Universal Network Enabled Analysis Kit (UNEAK) pipeline in TASSEL 3.0[51] was implemented due to its ability to distinguish heterozygous genotypes from paralogous loci in organisms lacking reference genome sequences such as *C. dactylon* in this study. Sequenced reads that matched the *Ape*KI enzyme cut site remnant (CWGC) and 96 barcodes were trimmed to 64 bp in length, and SNVs were called using the UNEAK pipeline[51]. Initially, SNVs were obtained with loose SNV calling criteria (minor allele frequency ≥ 0.01, missing data per site <90%). In order to reduce heterozygotes undercalling (i.e., incorrectly call a heterozygous genotype as homozygote due to low coverage), a minimum of six reads per SNV was imposed to call heterozygotes with a theoretical accuracy of 96.88% by assuming a binomial distribution of reads at two alleles for each locus[36,52]. Genotype data with each ≤5 reads were converted to missing data. Filtered SNVs and previously genotyped SSR markers[21] were assembled into a single dataset for genetic map construction.

**Linkage map construction**. To identify potential seed contaminants (i.e., non selfed progeny) in this population, we performed a principal coordinates analysis (PCoA) across 131 individuals (130 progeny and their parent) using SNV and SSR markers. PCoA was conducted in R (version 3.5.0; R Core Team, 2014) using *dist* and *cmdscale* functions. PCoA confirmed that these 130 progeny were truly selfed from A12359 (Supplementary Fig. 2). A linkage map was constructed with Join-Map 5[53]. A chi-square test was conducted on each SNV to test for the Mendelian segregation ratio of 1:2:1, and segregation distorted markers ($P < 0.01$) were excluded from the initial linkage group construction because distorted markers affect accuracy of linkage mapping by introducing errors in map distance estimation and marker ordering[54,55]. All markers were initially formatted to the <hk × hk> segregation type (i.e., heterozygous in the parent) according to JoinMap 5.0 user manual in Excel, and missing data were scored as "−". Markers were first

assigned to LGs based on a minimum log-likelihood of the odds (LOD) value of 10.0, and then the LOD threshold was decreased progressively to integrate ungrouped markers. A maximum likelihood mapping algorithm was used to order markers within each LG with the following parameters: linkages with a recombination frequency smaller than 0.4 and a LOD value larger than 1; goodness-of-fit jump threshold for removal of loci = 5; and number of added loci after which to perform a ripple = 1. Kosambi's mapping function[56] was used to calculate intermarker distance in centimorgans. To reduce redundant markers, one of each marker pair having an interval <0.4 cM was removed. The linkage map was constructed again using the same parameters except the maximum likelihood mapping algorithm was changed to a regression mapping algorithm. The LGs data were loaded to MapChart 2.32[57] to make the linkage group chart of the genetic map (Supplementary Data 3).

**Analysis of segregation distorted markers**. Segregation distortion is a very common phenomenon in plant species. To analyze segregation distortion, an additional genetic map was constructed without removing segregation distorted markers using the same method as described above except that more redundant markers were removed. A marker was excluded if the distance between it and the adjacent markers on an LG (made under the maximum likelihood mapping algorithm) was <1.5 cM. This genetic map, containing only the segregation distorted markers, was made with MapChart 2.32[57] and was compared with the previous one lacking severe segregation distorted markers.

**Comparative mapping between *C. dactylon* and other grasses**. *Cynodon dactylon* belongs to the subfamily Chloridoideae, which is composed of more than 1600 species, one of the largest grass subfamilies and is yet underexplored in terms of genomics and evolutionary studies[23,35]. To fill this gap and to provide further insight into the evolutionary history of grasses, we analyzed the genomic synteny and collinearity between the SNV-associated genomic sequences of *C. dactylon* and those of several other sequenced grass species. The 64 bp sequences of SNVs arranged on this *C. dactylon* genetic map were aligned against the reference genome sequences of *Oropetium thomaeum* v2.0[26], *Zoysia japonica* r1.1[58], *Sorghum bicolor* v2.0[43], *Setaria italic* v2.2[44], *Oryza sativa* v7.0[59], and *Miscanthus sinensis* v7.1 (DOE-JGI, http://phytozome.jgi.doe.gov/) (sequences of mapped SNVs used in comparison analyses given in Supplementary Data 1). *C. dactylon*, *O. thomaeum*, and *Z. japonica* belong to the subfamily Chloridoideae; *S. bicolor*, *S. italic*, and *M. sinensis* belong to the subfamily Panicoideae; *O. sativa* is a member of Ehrhartoideae (syn. Oryzoideae). Reference genome sequences of these grass species were downloaded from Phytozome v12.1 (https://phytozome.jgi.doe.gov/pz/portal.html) except for *O. thomaeum* v2.0, which was downloaded from CoGe (https://genomevolution.org/coge/), and *Z. japonica* r1.1, which was downloaded from Zoysia Genome Database (http://zoysia.kazusa.or.jp/index.html). Sequences of SNV markers were originally stored in the Tags on Physical Map file from the UNEAK pipeline, and were then formatted to the standard FASTA file using python program TagDigger[60]. Alignment was performed with Bowtie2[61] with stringent criteria: -D 20 -R 3 -N 1 -L 18 -i S,1,0.50–local. Genomic synteny plots between *C. dactylon* and the aforementioned six grass species were made in R (version 3.5.0; R Core Team, 2014) with customized scripts. Circos plots were generated using the circlize package[62].

**Statistics and reproducibility**. All statistical tests were performed using R (version 3.5.0) and base packages. Results are reproducible using the bioinformatic scripts and raw sequencing data that deposited on GitHub and NCBI, respectively.

**Reporting summary**. Further information on research design is available in the Nature Research Reporting Summary linked to this article.

## Data availability

Large datasets were generated and analyzed in this study. Three Illumina HiSeq 2500 runs generated ~225 GB of sequencing data. The data reported in this paper have been deposited in the Sequence Read Archive database, https://www.ncbi.nlm.nih.gov/sra (BioProject ID: PRJNA638432).

## Code availability

Bioinformatic scripts used in this study have been deposited on GitHub (https://github.com/hxdong-genetics/Bermudagrass_GBS_OSU).

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

## Acknowledgements

We would like to thank Pu Feng for maintaining the plant materials in a greenhouse. This research was funded in part by the USDA SCRI grant 2015-51181-24291.

## Author contributions

Y.Q.W. and J.F. conceived the research project; T.F., H.D. and S.Y. performed the research and data analyses; T.F., H.D., Y.Q.W., J.Q.M., C.H.F., D.L.M., J.F. wrote the paper.

## Competing interests

The authors declare no competing interests.
