## [Peer Review File · Communications Biology]

Reviewers' comments:

Reviewer #1 (Remarks to the Author):

This manuscript describes a study focused on genetic mapping of *Cynodon dactylon*. The authors made 18 linkage groups (LGs) of *C. dactylon* using genetic markers. They found that some LGs, which is orthologous to chromosome 10 of *Oropetium thomaeum*, were incorporated into two different lineage groups (LG3,4 and LG5,6). Merging chromosomes frequently occurs during evolution as the authors found, and so it is not so surprising. Most of their findings were just to confirm previous observations such as a history of p6-p9-p6 and p2-p10-p2 chromosomes using *C. dactylon*. This manuscript is descriptive. I did not find any significant advances bringing new biological insight in this study as in submission guideline. Thus I think that this work will not be publishable in *Communications Biology* even after revisions.

I found several typos.

Ancester
desseminated
asistanted
correpondence
chromsome
etc.

Reviewer #2 (Remarks to the Author):

The authors describe an investigation of the *Cynodon dactylon* genome based on genetic markers. This study provides additional genetic markers compared to two previous studies which are mentioned in the introduction. Genetic maps indicate how the chromosome number might have been reduced from 12 to 10 during the evolution of this lineage. A similar reduction through chromosome fusion was previously reported for other species. Despite some duplication between the results and method sections, the manuscript is generally well written with some minor mistakes (see comments below). I cannot find any major flaws.

General comments:

- 1) The authors might want to replace 'SNPs' by 'SNVs' as nothing is known about the function. These variants could have a pathogenic effect, but it is also possible that they are just polymorphisms.
- 2) Many technical details like parameters and tool names are included in the results section. The authors might want to move this information to the method section.
- 3)Code S1 and Code S2 missing.
- 4) line 342: Why were 100bp reads trimmed to 64bp? Barcodes should be substantially shorter than 30bp. The authors might want to comment on the quality of the reads e.g. by providing an average Phred score.
- 5) The authors might want to add versions of all bioinformatic tools (e.g. in line 343).

6) I was not able to find scripts in the supplements. There is a text document with some code though. The authors might want to consider uploading their scripts to a public repository like github.

Minor comments:

line 46: early genetic maps?

line 81 / line 82: "missing data per site <90%" and "missing rate more than 10%" seem to be redundant. The authors might want to check this.

line 150-line152: The authors might want to move this sentence to the discussion.

line 164-line165: This sentence might be better placed in the introduction.

line 173: genome > genome sequence ?

line 247: allotetraploid > allotetraploidy ?

line 278: ancestor > ancestor ? It seems that a word is missing in this sentence. The authors might want to rephrase it.

line 294: the the > the

line 300: genomes > genome sequences

line 328: "implemented" ?

One supplementary docx file (Table S1?) could be replaced by a text file which would facilitate re-use.

line 341: reference genome sequences / reference genome assemblies

line 348: The authors might want to check if a word is missing in the first sentence.

line 349: This seems to be redundant with a previous filter step. The authors might want to check.

line 390/395: reference genomes > reference genome sequences

Fig. 1: The authors might want to change the axes labels.

Reviewer #3 (Remarks to the Author):

This study investigated the genome evolution of Bermudagrass (*Cynodon dactylon*). The authors used 3,554 SNP and SSR markers from 130 S1 progeny to produce a genetic linkage map of this tetraploid genome. The linkage groups were then compared to the published genomes of other grasses to discern the chromosomal evolution of this species.

The introduction is very specific to *C. dactylon*. Further explanation of the broader impacts of this linkage map and the questions addressed would bolster the overall message.

The results are straightforward. Figure 1 (PCA of the S1) needs to be better explained and probably

should just be a supplementary figure as it is just confirming that no contaminant offspring made it into the analyses. The synteny plots can be combined into one or two figures rather than seven separate figures. Putting these analyses in a more phylogenetic context, rather than taxonomic context would be helpful. For example, a phylogeny of Poaceae with the divergence timing of the different subfamilies would be useful.

In terms of the methods, it seems to me that selecting only loci with 1:2:1 segregation ratios is not appropriate for any polysomic polyploid. The introduction and discussion suggest that it is still unclear if *C. dactylon* is disomic or polysomic.

Finally, there are numerous grammatical and spelling errors throughout the manuscript. In addition, the fully written scientific name of a species is often not provided before the abbreviated genus is used. Other abbreviations are not fully explained as in Lines 223-225 with the roman numerals. The genome sizes of these grasses should also be included alongside the chromosome numbers.

Note: reviewer's comments are in black color while author's responses in blue color.

Reviewer #1 (Remarks to the Author):

This manuscript describes a study focused on genetic mapping of *Cynodon dactylon*. The authors made 18 linkage groups (LGs) of *C. dactylon* using genetic markers. They found that some LGs, which is orthologous to chromosome 10 of *Oropetium thomaeum*, were incorporated into two different lineage groups (LG3,4 and LG5,6) Merging chromosomes frequently occurs during evolution as the authors found, and so it is not so surprising. Most of their findings were just to confirm previous observations such as a history of ρ_6 - ρ_9 - ρ_6 and ρ_2 - ρ_{10} - ρ_2 chromosomes using *C. dactylon*. This manuscript is descriptive. I did not find any significant advances bringing new biological insight in this study as in submission guideline. Thus I think that this work will not be publishable in Communications Biology even after revisions.

I found several typos.

Ancester

desseminated

asistanted

correpondence

chromsome

etc.

Author Response:

We agree with the reviewer that chromosome merging occurs frequently during evolution. In this study we found that paleo-chromosome 12 (syntenic to *Oropetium thomaeum* Chromosome 10) broke into two parts and each part segmentally disseminated into two homeologous LGs in *C. dactylon*, in addition to validating the previously reported ρ_6 - ρ_9 - ρ_6 and ρ_2 - ρ_{10} - ρ_2 chromosome merging in Poaceae. This finding is novel in the genome evolution of Poaceae. Moreover, bermudagrass is a widely used and economically important species but genomic information of bermudagrass is very limited. The dense genetic map constructed in this study provides a valuable tool to assist an ongoing whole genome sequence assembly and will facilitate genetic research such as marker assisted selection (MAS) in developing new cultivars.

All the typos mentioned have been corrected. We greatly appreciated the review.

Reviewer #2 (Remarks to the Author):

The authors describe an investigation of the *Cynodon dactylon* genome based on genetic markers. This study provides additional genetic markers compared to two previous studies which are mentioned in the introduction. Genetic maps indicate how the chromosome number might have been reduced from 12 to 10 during the evolution of this lineage. A similar reduction through chromosome fusion was previously reported for other species. Despite some duplication between the results and method sections, the manuscript is generally well written with some minor mistakes (see comments below). I cannot find any major flaws.

Author Response:

Thank you very much for your comments on our manuscript!

Please see below the response for each of your comments.

General comments:

1) The authors might want to replace 'SNPs' by 'SNVs' as nothing is known about the function. These variants could have a pathogenic effect, but it is also possible that they are just polymorphisms.

Response: SNPs were replaced with SNVs.

2) Many technical details like parameters and tool names are included in the results section. The authors might want to move this information to the method section.

Response: technical details were removed from the result section and incorporated into the method section.

3) Code S1 and Code S2 missing.

Response: we provided the code S1 and S2 in a file named “Supplementary information”.

4) line 342: Why were 100bp reads trimmed to 64bp? Barcodes should be substantially shorter than 30bp. The authors might want to comment on the quality of the reads e.g. by providing an average Phred score.

Response: We trimmed 100 bp raw reads to 64 bp in length because sequencing errors are enriched at the ends of reads. This trimming parameter is the default setting in UNEAK pipeline in TASSEL 3.0 (Lu et al., 2013), which we used.

Because reference genome sequence of *Cynodon dactylon* is not available, physical position of nucleotides cannot be obtained. UNEAK pipeline in TASSEL 3.0 also ignores quality scores, instead distinguishing alleles from errors based on how frequently they appear in the dataset. In

our analysis, we imposed a stringent criterion in filtering raw reads with depth < 6x, and this criterion should sufficiently remove low-quality reads.

Reference: Lu F, Lipka AE, Glaubitz J, Elshire R, Cherney JH, et al. (2013) Switchgrass genomic diversity, ploidy, and evolution: novel insights from a network-based SNP discovery protocol. *PLoS Genetics* 9(1): e1003215. doi:10.1371/journal.pgen.1003215

5) The authors might want to add versions of all bioinformatic tools (e.g. in line 343).

Response: We have added all bioinformatic tools we used.

6) I was not able to find scripts in the supplements. There is a text document with some code though. The authors might want to consider uploading their scripts to a public repository like github.

Response: All the files mentioned in the manuscript are provided in “Supplementary information”. As the scripts are specific to this manuscript, we believe it is appropriate just to provide them as supplementary in this manuscript, instead of put them to a public repository.

Minor comments:

line 46: early genetic maps?

Response: Changed to “two genetic maps”.

line 81 / line 82: "missing data per site <90%" and "missing rate more than 10%" seem to be redundant. The authors might want to check this.

Response: “missing data per site < 90%” was removed.

line 150-line152: The authors might want to move this sentence to the discussion.

Response: This sentence was removed.

line 164-line165: This sentence might be better placed in the introduction.

Response: The sentence was removed.

line 173: genome > genome sequence ?

Response: we changed it based on the suggestion.

line 247: allotetraploid > allotetraploidy ?

Response: We changed it.

line 278: ancestor > ancestor ? It seems that a word is missing in this sentence. The authors might

want to rephrase it.

Response: the sentence was changed to “It has been proposed that the common ancestor of grasses most likely had a basic chromosome number of seven.”

line 294: the the > the

Response: One ‘the’ was removed.

line 300: genomes > genome sequences

Response: we changed it.

line 328: "implemented" ?

Response: we changed it to “employed”.

One supplementary docx file (Table S1?) could be replaced by a text file which would facilitate re-use.

I remember Table S1 is an Excel file. Excel is very convenient and can be easily converted to text file using “save as”.

Response: We believe Excel file is appropriate here, so no conversion is made.

line 341: reference genome sequences / reference genome assemblies

Response: we changed to “reference genome sequences”

line 348: The authors might want to check if a word is missing in the first sentence.

Response: we changed to “Genotype data with each \leq five reads were converted to missing data.”

line 349: This seems to be redundant with a previous filter step. The authors might want to check.

Response: it was removed.

line 390/395: reference genomes > reference genome sequences

Response: we changed it.

Fig. 1: The authors might want to change the axes labels.

Response: We made changes to Fig. 1 based on the suggestion and moved it to a supplementary file as other reviewer suggested.

Reviewer #3 (Remarks to the Author):

This study investigated the genome evolution of Bermudagrass (*Cynodon dactylon*). The authors used 3,554 SNP and SSR markers from 130 S1 progeny to produce a genetic linkage map of this tetraploid genome. The linkage groups were then compared to the published genomes of other grasses to discern the chromosomal evolution of this species.

Response: thanks.

The introduction is very specific to *C. dactylon*. Further explanation of the broader impacts of this linkage map and the questions addressed would bolster the overall message.

Response: “The findings would add to the knowledge pool in the evolution of the grass family” was added by the end of the introduction section.

The results are straightforward. Figure 1 (PCA of the S1) needs to be better explained and probably should just be a supplementary figure as it is just confirming that no contaminant offspring made it into the analyses. The synteny plots can be combined into one or two figures rather than seven separate figures. Putting these analyses in a more phylogenetic context, rather than taxonomic context would be helpful. For example, a phylogeny of Poaceae with the divergence timing of the different subfamilies would be useful.

Response: Principal coordinates analysis (PCoA) was used to explore and to visualize similarities among selfed progeny and their parent ‘A12359’. The PCoA plot clearly showed the selfed progeny clustered around ‘A12359’, which confirmed the high-quality of population development and free of contamination. We have moved the Figure 1 to the supplemental files in this revised manuscript. We also combined the supplementary synteny plots into a single one.

In terms of the methods, it seems to me that selecting only loci with 1:2:1 segregation ratios is not appropriate for any polysomic polyploid. The introduction and discussion suggest that it is still unclear if *C. dactylon* is disomic or polysomic.

Response: It is critical to understand that we observed only two alleles at each SNV. We keep the markers with the 1:2:1 segregation ratio because the density of our map is very high, we want to keep less markers on the linkage map to avoid segregation distorted markers change the location of markers and cause problems for the genetic comparison. In addition, we also constructed genetic map by incorporating hundreds of segregation distorted markers.

Finally, there are numerous grammatical and spelling errors throughout the manuscript. In addition, the fully written scientific name of a species is often not provided before the abbreviated genus is used. Other abbreviations are not fully explained as in Lines 223-225 with the roman numerals. The genome sizes of these grasses should also be included alongside the

chromosome numbers.

Response: Grammatical and spelling errors in the manuscript were checked and corrected. We have added the full scientific name of species upon its first mention in abstract, main text, tables, and supplemental materials.

Changes made to the manuscript as follows:

1. Line 6, “the genome evolution” was changed to “genome evolution”
2. Line 7: "a" was changed to "the"
3. Line 10: "the" was added
4. Line 11: "a" was added
5. Line 14, “would facilitate” was changed to “facilitate”
6. Replaced “single nucleotide polymorphism” and ”SNPs” with “single nucleotide variant” and “SNVs”.
7. Line 29: removed a comma and added parentheses around *C. transvaalensis* Burtt-Davy
8. Line 30: "and" was changed to "or"
9. Line 32: "crop has" was changed to "crops have"
10. Line 43-44: "in *C. dactylon*" was moved from the end of sentence to middle for clarity
11. Line 44 "In one study," was added, “F1” was changed to “F₁”.
12. Line 48: "on the basis of" was changed to "based on"
13. Line 52: "also" was added
14. Line 52: "adding more" was changed to "who contributed additional"
15. Line 53: "The" was changed to "Their"
16. Line 55: comma added
17. Line 58: "one that derived from selfing 'A12359' has" was changed to "the one derived from selfing 'A12359' had"
18. Line 57-59: "Between the two S1 populations, one that derived from selfing of ‘A12359’ has less segregation distortion than that of ‘Zebra’, consequently the A12359 population was used for subsequent research including genetic mapping. Guo et al.²¹ reported an SSR marker-based genetic map in the A12359 population." was changed to "Between the two S1 populations, the one derived from selfing of ‘A12359’ had less segregation distortion than that derived from ‘Zebra’. Consequently, the A12359 population was

used for subsequent research including the development of an SSR marker-based genetic map²¹."

19. Line 64-66: "relatively small numbers of markers were mapped, specifically 291 loci in T89 and 252 loci in A12359" was changed to "a relatively small number of markers, specifically 291 loci in T89 and 252 loci in A12359, have been mapped."
20. Line 69: "of the present study" was added
21. Line 79, "were sequenced on an Illumina HiSeq 2500 platform" changed to "were sequenced with GBS"
22. Line 83: "and" was added.
23. Line 82: "and missing rate more than" was changed to "or a missing rate greater than", "and" was removed.
24. Line 85, "missing data per site < 90%" is removed.
25. Line 85: added "a"
26. Line 87: ""and 7,443 SNVs were retained at P>0.01" was changed to "and the 7,443 SNVs meeting this criterion at P>0.01 were retained"
27. Line 90, "Guo et al.²²" was changed to "Guo et al.²¹"
28. Line 91, "in JoinMap 5" is removed.
29. Line 101: added "The"
30. Line 103: added "the"
31. Line 117: removed "the"
32. Line 120: "Then the LGs were recalculated under regression algorithm with the remaining 2,904 markers, of which 1,861 (64.08%) were distorted markers and mapped on the 18 LGs" was changed to "Using these 2,904 markers, of which 1,861 (64.08%) were distorted markers, we recalculated the 18 LGs under the regression algorithm (Figure 2)."
33. Line 126: "almost all" was changed to "each of"
34. Line 130: "was" was added
35. Line 142: removed comma and added "an"
36. Line 155: removed comma

37. Line 156, “Although not unexpected, such phenomena indicated that rearrangements of local ancestral chromosomal segments frequently occurred during the *C. dactylon* evolution history.” was removed.
38. Line 166, “compared” was moved from before “these four LGs” to after and added “with”.
39. Line 168, added comma
40. Line 169, “*Zoysia japonica* is a perennial grass belonging to the genus of *Zoysia* in the subfamily Chloridoideae. Both *Z. japonica* and *C. dactylon* are widely used as turfgrass in home lawns, sports fields and golf courses, and for forage as well.” was removed.
41. Line 175: "corresponds" was changed to "corresponded"
42. Line 186-187, "corresponds" was changed to "corresponded"
43. Line 189: added "and"
44. Line 207: "intermarker spacing was 0.41cM in the map while those values of the Khanal et al." was changed to "intermarker spacing in this map was 0.41cM, while those values from Khanal et al."
45. Line 211, changed “Y. Wu” to “Y.Q. Wu”
46. Line 220-221: added comma
47. Line 232: removed "bermudagrass"
48. Line 242 and Line 244, “are” was changed to “were”
49. Line 243: comma added
50. Line 249-253: “allotetraploid” was changed to “allotetraploidy”
51. Line 250: this sentence was split into two sentences
52. Line 258: added comma
53. Line 264-266: "we compared *C. dactylon* with *O. thomaeum* and *Z. japonica*, which are members of subfamily Chloridoideae; with *S. bicolor*, *S. italica*, and *M. sinensis*, members of Panicoideae; with *O. sativa*, a member of Ehrhartoideae" was changed to "we compared *C. dactylon* with *O. thomaeum* and *Z. japonica* of the subfamily Chloridoideae; *S. bicolor*, *S. italica*, and *M. sinensis* of Panicoideae; and *O. sativa* of Ehrhartoideae"
54. Line 270, “Therefore, it is not unexpected that *C. dactylon* showed similarly highest genomic syntenic relationships with *O. thomaeum* and *Z. japonica* due to their close phylogenetic relationship within the Chloridoideae subfamily (Table 2).” was changed to

“Therefore, it is not unexpected that *C. dactylon* showed high genomic syntenic relationships with *O. thomaeum* and *Z. japonica* due to their close phylogenetic relationship within the Chloridoideae subfamily (Table 2).”

55. Line 273: added comma

56. Line 273, 276, “agree” was changed to “agreed”

57. Line 287: added "have"

58. Line 290, “chromosome of seven” was changed to “chromosome number of seven”

59. Line 301, “involve” was changed to “be engaged”

60. Line 303, “to involve” was removed

61. Line 304, “the” removed

62. Line 306: “the the” was changed to “the”

63. Line 312-313: “genomes” was changed to “genome sequences”.

64. Line 321-327, “that is” changed to “corresponding to”, add “and” and deleted “which corresponds to”, removed “comma”, added “to have occurred”

65. Line 330: added "the"

66. Line 342: "implemented" was changed to “employed”.

67. Line 348-350, “each of the 130 progeny” was deleted, “sequencing” was deleted, “each progeny” was added, “14 times for” was added before the parent, and “DNA was replicated 14 times,” was removed, “which was” was added, “the” was deleted

68. Line 356: “reference genome” was changed to “reference genome sequences”

69. Line 358: added comma

70. Line 363: added "for"

71. Line 363-364: changed to “Genotype data with \leq five read repeats were converted to missing data.”

72. Line 364: “Then SNPs with missing rate no more than 10% were retained.” Was removed.

73. Line 367: changed “could be found from" to "can be found in"

74. Line 373: added "A" to both sentences

75. Line 380: added "the"

76. Line 381: added "A"

77. Line 384: added "and"

78. Line 386: "one of marker pairs with an interval less than 0.4 cM" was changed to "one of each marker pair having an interval less than 0.4 cM"
79. Line 388: "a" was added
80. Line 389: "was" changed to "were"
81. Line 375: "more redundant markers" was changed to "a greater number of redundant markers"
82. Line 376: "The" was changed to "A"
83. Line 376: "marker was excluded if it is less than 1.5 cM to the adjacent markers on the LGs made under maximum likelihood mapping algorithm." was changed to "marker was excluded if the distance between it and the adjacent markers on an LG (made under the maximum likelihood mapping algorithm) was less than 1.5 cM."
84. Line 398: "without" was changed to "lacking"
85. Line 407: "reference genome" was changed to "reference genome sequences"
86. The previous Figure 1 was changed to Figure S1, the previous figures 2-5 were changed to Figures 1-4 accordingly, the previous figures S1-S5 were changed to figures S2A-E accordingly.
87. Code S1 and Code S2 were included.
88. References were re-proofed and corrected the inaccurate numbering

Reviewers' comments:

Reviewer #2 (Remarks to the Author):

The authors addressed some of my comments, but there are also issues listed below. The overall character of the manuscript remains descriptive without providing mechanistic insights for the chromosome evolution. Nevertheless, I think this work can be turned into a technically solid validation of previous findings in another species.

1) ok

2) There are still technical details in the results section e.g. line 80 "using the UNEAK pipeline in TASSEL". This should be moved to the method section. In addition, the authors should carefully check if an appropriate reference is included for all tools.

3) Yes, but these files are now a PDF. Re-use/validation would require extraction of code from this PDF. Even if the code is only useful to repeat the presented analysis this should be supported. Therefore, the authors should place their code in a public repository like github if submission as scripts in the supplements is not possible.

4) Additional details should be provided in the method section. Trimming based on the actual quality score of reads could result in a substantially increased specificity. Unless the sequencing quality is extremely low, more than 64 bases of each read should be of sufficient quality. The authors might want to provide an analysis of the initial and the trimmed read quality distribution to justify the use of a default value. The identification of a comprehensive set of high quality SNVs is crucial for this study.

5) ok

6) See comment 3

Reviewer #3 (Remarks to the Author):

The authors have cleaned up the text considerably, however, this study still lacks a major conclusion. The focus seems to be the evolution of these chromosomes across the sampled grasses using one on one synteny analyses with *C. dactylon*. If the authors can show the evolution of their chromosomes of interest in a phylogenetic context, rather than A vs B, I think the message will be much more succinct. As is, I agree with the first reviewer that this is a largely descriptive study with little biological insight.

Line 46: "single doze" should be "single dose"

Reviewers' comments in black color and responses in blue color.

Reviewer #2 (Remarks to the Author):

The authors addressed some of my comments, but there are also issues listed below. The overall character of the manuscript remains descriptive without providing mechanistic insights for the chromosome evolution. Nevertheless, I think this work can be turned into a technically solid validation of previous findings in another species.

1) ok

2) There are still technical details in the results section e.g. line 80 "using the UNEAK pipeline in TASSEL". This should be moved to the method section. In addition, the authors should carefully check if an appropriate reference is included for all tools.

Response: "by using the UNEAK pipeline in TASSEL" was removed. We have double checked the references for each tool we used.

3) Yes, but these files are now a PDF. Re-use/validation would require extraction of code from this PDF. Even if the code is only useful to repeat the presented analysis this should be supported. Therefore, the authors should place their code in a public repository like github if submission as scripts in the supplements is not possible.

Response: we have deposited all bioinformatic scripts in github (https://github.com/hxdong-genetics/Bermudagrass_GBS_OSU). We also added one sentence in the Methods section to clarify this.

4) Additional details should be provided in the method section. Trimming based on the actual quality score of reads could result in a substantially increased specificity. Unless the sequencing quality is extremely low, more than 64 bases of each read should be of sufficient quality. The authors might want to provide an analysis of the initial and the trimmed read quality distribution to justify the use of a default value. The identification of a comprehensive set of high quality SNVs is crucial for this study.

Response: We evaluated the base quality of raw sequencing data using FastQC v0.11.7 (<https://www.bioinformatics.babraham.ac.uk/projects/fastqc/>). All three GBS libraries were sequenced with high-quality reads, and only two ends of reads showed slightly lower quality score. Therefore, we trimmed both ends of reads to reduce error-prone data. We provided the new results of our base quality analysis in the supplemental materials (Figure S1) and added brief description of sequencing raw data quality in the revised manuscript. We agree with the reviewer that longer reads are desirable in genomic analysis. By trimming raw reads to 64-bp in length, we focused on the highest-quality SNVs and should be of great power in

capturing the genetic variation in this population, especially this is a first-generation selfed population and linkage disequilibrium remains extensive.

5) ok

6) See comment 3

Response: The previous Code S2 is actually the input file for MapChart software. It was changed to "Table S3. Input file for MapChart for segregation distorted markers on linkage groups (LGs) 1 to 18. 3-1, 4-1 5-1 and 6-1."

Reviewer #3 (Remarks to the Author):

The authors have cleaned up the text considerably, however, this study still lacks a major conclusion. The focus seems to be the evolution of these chromosomes across the sampled grasses using one on one synteny analyses with *C. dactylon*. If the authors can show the evolution of their chromosomes of interest in a phylogenetic context, rather than A vs B, I think the message will be much more succinct. As is, I agree with the first reviewer that this is a largely descriptive study with little biological insight.

Response: We added Figure 5 and a conclusion. In Figure 5, we demonstrated the chromosome number reduction of sequenced and fine mapped grasses within Panicoideae and Chloridoideae subfamilies

Line 46: "single doze" should be "single dose"

Response: "single doze" was changed to "single dose".

Additional changes made in this revision

1. Line 115, changed 'larger than' to 'less than'.
2. Line 134, changed 'Figure S2A' to 'Figure S3A'.
3. Line 161, added ',' after 'As expected'
4. Line 162, changed 'Figure S2' to 'Figure S3'
5. Line 164, changed 'Figure S2' to 'Figure S3'
6. Line 174-175, changed 'Figure S2' to 'Figure S3'
7. Line 176, changed 'Figure S2' to 'Figure S3'
8. Line 178, changed '2' to '3'
9. Line 273, changed 'Figure S2' to 'Figure S3'

10. Line 299, changed 'Figure S2' to 'Figure S3'
11. Line 315-318, changed 'one half of rice chromosome 12 inserted into 6, ρ6-ρ12-ρ6) and LGs 5, 6 (the other half of rice chromosome 12 into 1)' to 'one half of paleo-ancestor chromosome 12 inserted into chromosome 6, ρ6-ρ12-ρ6) and LGs 5, 6 (the other half of paleo-ancestor chromosome 12 inserted into 1, ρ1-ρ12-ρ1)'
12. Line 320-324, added one sentence 'To summarize the results in a phylogenetic context, the evolutionary model to demonstrate the chromosome number reduction for *C. dactylon* was modified from previous publications^{36, 62} of sequenced and mapped grasses in the Panicoideae, Chloridoideae, and Ehrhardoideae subfamilies (Figure 5).'
13. Line 374, changed 'Figure S1' to 'Figure S2'
14. Line 397-398, changed 'distortion' to 'distorted'
15. Line 407, add 'of' before 'Oropetium'
16. Line 432, changed 'Figure S1' to 'Figure S2', 'coordinate' to 'coordinates'
17. Line 434-439, changed 'S2' to 'S3', added 'Comparative mapping between *Cynodon dactylon* and other grass species', changed '*Cynodon*' to '*C.*', added (LGs) after linkage groups, and changed 'linkage groups' to 'LGs'
18. Line 576, added one more reference

REVIEWERS' COMMENTS:

Reviewer #2 (Remarks to the Author):

2) ok

3) ok, but a README (TXT file) should be included.

4) The authors assessed the read quality and revealed that a more lenient trimming approach would be possible. The quality of detected variants should increase, because longer reads can be aligned with higher accuracy. The authors might want to check

a) if the overall results of their study are affected by a variant calling based on this alternative trimming approach

b) if the transition to transversion ratio of the identified SNVs differs between both approaches. This can indicate which variant calling set has the higher quality.

Reviewer #3 (Remarks to the Author):

The ms is improved with the additional chromosome evolution discussion and figure. My only recommendation would be to switch common names for scientific in the figure and text.

REVIEWERS' COMMENTS:

Reviewer #2 (Remarks to the Author):

2) ok

3) ok, but a README (TXT file) should be included.

Response: the "README" (TXT file) was added, all the descriptions of the code files were included.

4) The authors assessed the read quality and revealed that a more lenient trimming approach would be possible. The quality of detected variants should increase, because longer reads can be aligned with higher accuracy. The authors might want to check

a) if the overall results of their study are affected by a variant calling based on this alternative trimming approach

b) if the transition to transversion ratio of the identified SNVs differs between both approaches. This can indicate which variant calling set has the higher quality.

Response: Thank the reviewer very much for the suggestion. Your comments are highly valued and appreciated. We understand that the longer reads would increase the accuracy for the alignment, but we would prefer to keep the read length at 64 bps because: (1) 64 bps is long enough to balance the reads quality and reads length, (2) the chance of two or more SNVs appear within ~100 bps sequence is very small, (3) keeping longer length of the reads would introduce more low quality sequences and would decrease the SNV calling number.

We don't anticipate a big change of the linkage groups and comparative genetics results if the length of the reads would be changed because of the good quality of our reads, the high depth of sequencing, and the high density of our linkage groups. But, but that would result in repeating all our analyses from the beginning all over again. This would be a good reminder for us to do SNP calling in the future though.

Reviewer #3 (Remarks to the Author):

The ms is improved with the additional chromosome evolution discussion and figure. My only recommendation would be to switch common names for scientific in the figure and text.

Response: We changed the common names to scientific names of the species in Fig. 5 and made it uniform with the text.

Line 276: "Rice" was changed to "*O. sativa*"

Additional changes

1. Line 3-11, authors and affiliations were included.
2. Line 14-25, sentences were modified in present tense.

3. Line 22-24, the sentence was modified as 'A segmental dissemination of the paleo-chromosome p12 (p1-p12-p1, p6-p12-p6) leads to the 10-to-9 chromosome reduction in *C. dactylon* genome.'
4. 'Figure' was changed to 'Fig.' throughout the manuscript.
5. 'Table S' was changed to 'Supplementary Table' throughout the manuscript.
6. 'Figure S' was changed to 'Supplementary Fig.'
7. Subheadings in DISCUSSION were removed.
8. Line 270, 'Bambusoideae' was changed to 'Ehrhartoideae'.
9. Line 331, 'CONCLUSIONS' was removed.
10. Line 352-353, 'as described previously' was removed.
11. Line 373-374, 'Detailed scripts of SNV calling and filtering can be found in Supplementary Data 1.' was deleted as we have deposited the scripts at GitHub.
12. Line 431-454, 'Statistics and Reproducibility', 'Data availability', and 'Reporting summary' were added while 'SUPPLEMENTARY MATERIALS' sections were deleted.
13. Page 26, 'Acknowledgements', 'Author contributions', 'Competing interests', and 'Code availability' were included.